# Effects of Angiotensin II on Erythropoietin Production in the Kidney and Liver

**DOI:** 10.3390/molecules26175399

**Published:** 2021-09-05

**Authors:** Yukiko Yasuoka, Yuichiro Izumi, Takashi Fukuyama, Hideki Inoue, Tomomi Oshima, Taiga Yamazaki, Takayuki Uematsu, Noritada Kobayashi, Yoshitaka Shimada, Yasushi Nagaba, Masashi Mukoyama, Yuichi Sato, Jeff M Sands, Katsumasa Kawahara, Hiroshi Nonoguchi

**Affiliations:** 1Department of Physiology, Kitasato University School of Medicine, 1-15-1 Kitasato, Minami-ku, Sagamihara 252-0374, Kanagawa, Japan; yasuoka@med.kitasato-u.ac.jp (Y.Y.); tomomio@kitasato-u.ac.jp (T.O.); kawahara@kitasato-u.ac.jp (K.K.); 2Department of Nephrology, Graduate School of Medical Sciences, Kumamoto University, 1-1-1 Honjo, Chuo-ku, Kumamoto 860-8556, Kumamoto, Japan; izumi_yu@kumamoto-u.ac.jp (Y.I.); hideki24@kumamoto-u.ac.jp (H.I.); mmuko@kumamoto-u.ac.jp (M.M.); 3Division of Biomedical Research, Kitasato University Medical Center, 6-100 Arai, Kitamoto 364-8501, Saitama, Japan; fukuyam@insti.kitasato-u.ac.jp (T.F.); tyamazak@insti.kitasato-u.ac.jp (T.Y.); tuematsu@insti.kitasato-u.ac.jp (T.U.); kenchu@insti.kitasato-u.ac.jp (N.K.); 4Division of Internal Medicine, Kitasato University Medical Center, 6-100 Arai, Kitamoto 364-8501, Saitama, Japan; yoshi@insti.kitasato-u.ac.jp (Y.S.); nagaba-y@insti.kitasato-u.ac.jp (Y.N.); 5Department of Molecular Diagnostics, School of Allied Health Sciences, Kitasato University, 1-15-1 Kitasato, Minami-ku, Sagamihara 252-0373, Kanagawa, Japan; yuichi@med.kitasato-u.ac.jp; 6Renal Division, Department of Medicine, Emory University School of Medicine, 1639 Pierce Drive, WMB Room 3313, Atlanta, GA 30322, USA; jeff.sands@emory.edu

**Keywords:** erythropoietin, angiotensin II, renin–angiotensin–aldosterone system, deglycosylation, Western blotting, HIF2α, proximal tubules, collecting ducts

## Abstract

The kidney is a main site of erythropoietin production in the body. We developed a new method for the detection of Epo protein by deglycosylation-coupled Western blotting. Detection of deglycosylated Epo enables the examination of small changes in Epo production. Using this method, we investigated the effects of angiotensin II (ATII) on Epo production in the kidney. ATII stimulated the plasma Epo concentration; Epo, HIF2α, and PHD2 mRNA expression in nephron segments in the renal cortex and outer medulla; and Epo protein expression in the renal cortex. In situ hybridization and immunohistochemistry revealed that ATII stimulates Epo mRNA and protein expression not only in proximal tubules but also in collecting ducts, especially in intercalated cells. These data support the regulation of Epo production in the kidney by the renin–angiotensin–aldosterone system (RAS).

## 1. Introduction

Erythropoietin (Epo) stimulates erythrocyte production in the bone marrow [1]. Hypoxia and anemia stimulate Epo production by the kidney [1,2,3,4]. Severe hypoxia and severe anemia cause more than a 500-fold increase in the serum Epo concentration [5]. However, such conditions are rare and most people do not experience them during their lives. The serum Epo concentration in normal people is low, and Epo protein expression in control kidneys has been too low to detect [5,6,7,8]. We developed a new Western blotting method and found that deglycosylated Epo is an accurate marker of Epo expression [5,9]. Epo protein in the kidney is detected at 34–43 kDa, and deglycosylation using PNGase shifts the band to 22 kDa. The broad Eoo band is concentrated to a narrow 22 kDa band, which results in an increase in detection sensitivity by 10–100-fold [5,9]. Thus, the detection of small changes in Epo production has become possible.

The renin–angiotensin–aldosterone system (RAS) regulates renal water and electrolyte excretion to maintain body fluid homeostasis [10,11]. RAS inhibitors cause anemia by reducing Epo production. Kim et al. reported that angiotensin II (ATII) increases Epo production via angiotensin 1 receptor (AT1R)-mediated early growth response-1 (Egr-1) activation in human renal 786-O cells [12,13] Kato et al. reported that transgenic mice having renin and angiotensinogen genes exhibit erythrocytosis and that knockout mice lacking renin and angiotensinogen exhibit anemia [14,15]. The anemia of angiotensinogen-knockout mice was rescued by ATII infusion, suggesting that a genetic pathway from angiotensinogen to ATII plays a key role in ATII-induced Epo production. These reports suggest that Epo production is regulated not only by hypoxia or anemia but also by RAS [4,12,14,15,16,17,18]. In our previous reports, we showed that aldosterone stimulates Epo production by the intercalated cells of the collecting ducts [7]. In this report, we examined the effects of ATII on Epo production by the kidney and liver.

## 2. Results

### 2.1. Plasma Epo Concentration

ATII administration increased the plasma Epo concentration from 1.4 ± 0.2 to 2.9 ± 0.3 mIU/mL (Figure 1a).

### 2.2. Epo mRNA Expression in the Renal Cortex and Microdissected Proximal Convoluted Tubules (PCT) and Medullary Thick Ascending Limbs (MAL)

ATII stimulated Epo and HIF2α mRNA expression in the renal cortex by 13.2 ± 4.7 and 11.4 ± 0.6 times by A1 and A10, respectively (*n* = 3–4) and 9.6 ± 3.7 and 8.6 ± 2.6 times by A1 and A10, respectively (*n* = 3–4) (Figure 1b,c). In microdissected PCT, ATII increased Epo mRNA by 3.4 ± 0.7 and 2.4 ± 0.2 times by 10^−9^ M and 10^−6^ M ATII, respectively (*n* = 3) and HIF2α mRNA by 2.0 ± 0.2 and 2.3 ± 0.8 times by 10^−9^ M and 10^−6^ M ATII, respectively (*n* = 3) (Figure 1d,e). ATII also increased Epo mRNA expression in MAL by 4.3 ± 1.0 and 6.5 ± 2.0 times by 10^−9^ M and 10^−6^ M ATII, respectively (*n* = 3) (Figure 1f).

### 2.3. In Situ Hybridization of Epoand the Effects of ATII

Epo mRNA expression was observed in cortical nephrons under basal conditions, as previously reported [19]. ATII stimulated Epo mRNA expression in PCT, MAL, and cortical and medullary collecting ducts (CCD and OMCD) after 2 h (Figure 2: 2a, 2d). HIF2α mRNA expression increased in most nephron segments and peritubular cells of both the cortex and the medulla (Figure 2: 2–4b, 2–4e). PHD2 mRNA expression also increased in MAL, CCD, and OMCD (Figure 2: 3–4c, 3–4f). ATII failed to induce Epo mRNA expression in the peritubular cells of the renal cortex.

### 2.4. Western Blotting Analysis of ATII-Induced Epo Production in the Kidney and Liver

The effects of fludrocortisone, an aldosterone agonist, were examined using deglycosylation-coupled Western blotting. The band of deglycosylated Epo at 22 kDa was observed in control rats (C2, C4, and C6 in Figure 3a right side), showing that Epo is produced in control rats. Fludrocortisone significantly stimulated the expression of deglycosylated Epo at 2 and 4 h after injection. The expression became equal to that in control rats after 6 h. Next, the effect of ATII on Epo expression was examined. The effect of ATII on Epo production was larger in the renal cortex than in the renal outer medulla (Figure 3b). The effect of ATII on Epo production was not observed in the renal inner medulla (data not shown). ATII dose-dependently increased Epo protein expression (Figure 3c). ATII increased deglycosylated Epo production in the renal cortex by 1.6 ± 0.2 and 3.4 ± 0.3 times by 1 and 10 mg/kg, respectively (*n* = 6–8). In contrast, Epo production was not seen in the liver, even after the administration of ATII (Figure 3d).

### 2.5. Immunohistochemistry of Epo Production by the Kidney

By IHC of mice, under basal conditions, Epo staining was found from the proximal tubules to the collecting ducts (proximal tubules < thick ascending limbs < collecting ducts) (Figure 4). ATII increased Epo production in PCT, MAL, CCD, and OMCD after 4–6 h (Figure 4; 3a, 3b, 3c, 4a, 4b and 4c). In the collecting duct, Epo expression was detected only in the type A intercalated cells, as previously reported (brown cells in Figure 4; 3c and 4c) [7,19]. Epo staining was not detected in peritubular cells under these conditions.

## 3. Discussion

The detection of Epo by Western blotting was difficult until our finding of a deglycosylation-coupled assay [5,7,9,20,21,22,23,24,25,26]. In severe hypoxia and anemia, the production of Epo is stimulated to a high level and the detection of Epo is easy [2,3,4,5]. However, such conditions are rare in vivo. We found that Epo is produced by the kidney even under control conditions. The detection of Epo in the control kidney is difficult, even by our Western blotting. Deglycosylation of Epo by PNGase increases the band intensity of Epo at 22 kDa, making it possible to detect lower levels of Epo. Using this method, we investigated the effects of fludrocortisone and ATII on Epo production.

ATII increased Epo mRNA expression in the renal cortex. Next, we examined the effects of ATII on Epo mRNA expression in microdissected PCT and MAL. ATII increased Epo mRNA expression in PCT and MAL. We also examined the effects of ATII by in situ hybridization (ISH). ISH revealed that ATII increases Epo mRNA expression not only in proximal tubules but also in distal tubules. Furthermore, we investigated Epo protein expression by Western blotting. Deglycosylation-coupled Western blotting was used in this study. The results show that ATII dose-dependently increases Epo protein expression in the kidney but not in the liver. The plasma Epo concentration increased by 2–3-fold, which is compatible with previous reports [4,6,8,12]. This level of Epo production in blood cannot be measured by Western blotting. However, deglycosylation-coupled Western blotting made the detection possible. ATII significantly increased Epo protein expression in the kidney. Immunohistochemistry also showed that ATII increases Epo protein expression in both proximal and distal tubules. The effect of ATII is identical to that of fludrocortisone in the intercalated cells of the collecting ducts [7]. Our study is the first one that shows the intrarenal localization of ATII-induced Epo production by ISH and IHC.

ATII also increased HIF2 mRNA expression in the renal cortex and proximal convoluted tubules, showing that the increase in Epo production by ATII is caused by the stimulation of HIF2, the same as aldosterone [7]. Our study indicates that Epo production by the kidney is regulated by the renin–angiotensin–aldosterone system. The role of ATII on Epo production in the kidney has been suggested [4,12,16,17,18]. Jelkmann suggested indirect effects of angiotensin II as a growth factor for myeloid erythrocytic progenitors [4]. Kim et al. clearly reported that angiotensin II stimulates Epo production via AT1R-mediated Egr-1 activation by p21Ras-mitogen-activated protein kinase/extracellular signal-regulated kinase (ERK) kinase-ERK/2 in human renal 786-O cells [12,13]. The role of a genetic pathway from angiotensinogen to ATII was examined by Kato et al. [14,15]. They showed that transgenic mice having renin and angiotensinogen genes exhibit erythrocytosis. They also reported that knockout mice lacking renin and angiotensinogen exhibit anemia that was rescued by ATII infusion, suggesting that a genetic pathway from angiotensinogen to ATII plays a key role in ATII-induced Epo production. The mechanisms of Epo production under severe hypoxic and basal conditions may be different, but some steps such as the stimulation of HIF2 could be the same. The ability to produce Epo would be different between renal interstitial cells and nephrons. The detection of HIF2 by Western blotting was not successful in this study; we could not detect the HIF2 band at around 115 kDa, even by the use of sc-46691, sc-13596, Ab8365, and NB100-122.

## 4. Materials and Methods

### 4.1. Materials and Animals

Male Sprague–Dawley rats (Japan SLC, Hamamatsu, Japan) and C57BL/6J mice (Charles River Japan, Yokohama, Japan) were used. Rats were given 1 or 10 mg/kg of ATII (A1 and A10, respectively), and mice were given 5 mg/kg of ATII by intraperitoneal injection. The control group was injected with saline. After 4 h, rats were injected with a mixed anesthetic (0.3 mg/kg of medetomidine, 4.0 mg/kg of midazolam, and 5.0 mg/kg of butorphanol), and blood was taken from the abdominal aorta. The kidneys and liver were dissected after perfusing 20 mL of PBS into the abdominal aorta. In some experiments, kidney slices were incubated in collagenase (type 1, C-0130; Sigma-Aldrich, Burlington, NJ, USA) and ribonucleoside vanadyl complexes (R-3380; Sigma-Aldrich) for 20 min and PCT and MAL were microdissected, as described previously [7]. Microdissected PCT or MAL were incubated with ATII or the vehicle for 2 h at 37 °C. Mice were anaesthetized with 1.5% isoflurane in 30% O_2_ (a mixture of 100% O_2_ and air) after 2, 4, and 6 h. The kidneys were quickly removed and cut longitudinally. The main kidney pieces were fixed by immersion in ice-cold 4% paraformaldehyde/0.1 M phosphate buffer overnight and processed in paraffin for histological analysis.

Our protocols were checked and approved by the Institutional Ethics Committee of the Kitasato University Medical Center (2018032, 2019029) and the Kitasato University School of Medicine (2019–141).

### 4.2. Real-Time PCR

RNA was extracted from the kidney, liver and microdissected PCT or MAL using Qiacube and the RNeasy Mini Kit (74106; Qiagen, Venlo, The Netherlands), as described previously [5]. cDNA was synthesized using a Takara PrimeScript II 1st strand cDNA Synthesis Kit (6210; Takara, Otsu, Japan). Real-time PCR was performed using probes from Applied Biosystems (GAPDH Rn01775763_g1, Epo Rn00667869_m1, HIF2 Rn00576515_m1, HIF1a Rn01472831_m1, PHD2 Rn00710295_m1; Waltham, MA, USA) and Premix Ex Taq (RP39LR; Takara). mRNA expression in control and ATII rats was compared using ΔΔCT.

### 4.3. In Situ Hybridization

ISH was performed, as described previously [5,27]. In brief, total RNA from the mouse kidneys (636612; BD Bioscieces Clontech) was reverse-transcribed with an RNA PCR kit (AMV), ver. 3.0 (RR019; Takara), and the cRNA probe for Epo (GenBank accession no. NM_007942), HIF2α (GenBank accession no. NM_010137), or PHD2 (GenBank accession no. NM_053207) was generated with T7 promoter region-tailed PCR primers. The hybridized sections were successively treated with 0.1% avidin, 0.01% biotin solution, 0.5% casein/TBS, horseradish peroxidase (HRP)-conjugated sheep anti-DIG F (ab′) fragment antibody (11207733910; Roche Diagnostics, Basel, Switzerland), biotinylated tyramide solution, and HRP-conjugated streptavidin (P0397; DakoCytomation, Glostrup, Denmark). Sections were stained using the DAB liquid system (BSB 0016; Bio SB, Santa Barbara, CA, USA) and Mayer’s hematoxylin (30002; Muto Pure Chemicals, Tokyo, Japan).

### 4.4. Immunohistochemistry

Kidney sections were immuno-stained, as described previously [7,28]. In brief, the sections were blocked with 5% normal goat serum and reacted with rabbit polyclonal anti-human Epo antibody (sc-7956, 1:50; Santa Cruz Biotechnology, Santa Cruz, CA, USA), followed by Histofine Simple Stain MAX-PO (414341F; Nichirei Bioscience, Tokyo, Japan). Sections were stained using the DAB liquid system and counterstained with Mayer’s hematoxylin.

Images were obtained using an optical microscope (Axioplan 2; Carl Zeiss, oberkochen, Germany) with a digital camera (AxioCam MRc5; Carl Zeiss). Captured images were analyzed using an image analysis system (AxioVision Rel. 4.6; Carl Zeiss).

### 4.5. Western Blotting Analysis with Deglycosylation

Western blotting analysis was performed, as described previously [4,6,7]. Protein was collected from the renal cortex and liver using CelLytic MT (C-3228; Sigma-Aldrich) and used for Western blotting. Kidney and liver samples were deglycosylated using N-glycosidase F (PNGase, 4450; Takara, as described previously. In brief, 1 μL of 10% SDS was added to 10 μL of kidney or liver samples and boiled for 3 min. Then, 11 μL of 2× stabilizing buffer was added and the samples were vortexed. After the addition of 1–2 μL of PBS or PNGase, the samples were incubated in a water bath for 20 h at 37 °C. After incubation, the samples were spun down and the supernatant was collected and used for SDS-PAGE (10–20% gradient gel; Cosmo Bio No. 414893, Tokyo, Japan). The 2× stabilizing buffer contained 125 mM Tris-HCl (pH 8.6), 48 mM EDTA, 4% Nonidet P-40, and 8% 2-mercaptoethanol. Recombinant human Epo (rhEpo, 587102; BioLegend, San Diego, CA, USA) was used as a positive control. After SDS-PAGE, proteins were transferred to a PVDF membrane (Immobilon-P; IPVH00010; Merck Millipore, Burlington, NJ, USA) with 120 mA for 60–90 min. The membrane was blocked with 5% skim milk (Morinaga, Japan) for 60 min and incubated with the antibody against Epo (sc-5290, 1:500; Santa Cruz Biotechnology) for 60 min at room temperature. After washing, the membrane was incubated with a secondary antibody (goat anti-mouse IgG (H + L); 115–035-166, 1:5000; Jackson Immuno Research Laboratories, West Grove, PA, USA) for 60 min. Bands were visualized by the ECL Select Western Blotting Detection System (RPN2235; GE Healthcare Bio-Science AB, Chicago, IL, USA) and LAS 4000 (Fujifilm, Tokyo, Japan).

### 4.6. Plasma Epo Concentration Measurements

Plasma was collected from control and ATII-treated rats at 4 h after peritoneal injection. Plasma Epo concentrations were measured by CLEIA (SRL, Tokyo, Japan) using Access Epo by Beckman Coulter (Brea, CA, USA).

### 4.7. Statistical Analyses

Data are expressed as the mean ± SEM. Statistical significance was performed using Excel Statistics (BellCurve, Tokyo, Japan). Statistical significance was analyzed using a *t*-test or a non-parametric analysis using the Kruskal–Wallis test and multiple comparisons by the Shirley–Williams’ test. *p*-Values < 0.05 were considered statistically significant.

## 5. Conclusions

In conclusion, our study shows that ATII increases Epo mRNA and protein expression in proximal and distal renal tubules. Combined with our previous studies, Epo production by the kidney under basal conditions is regulated by the renin–angiotensin–aldosterone system.

## Figures and Tables

**Figure 1 molecules-26-05399-f001:**
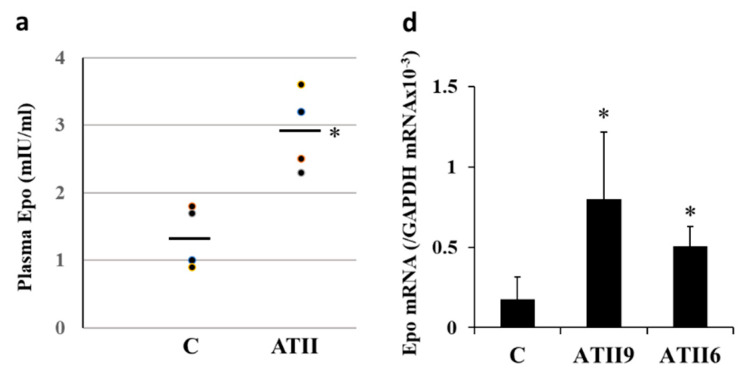
Epo mRNA expression. (**a**) Plasma Epo concentration in control and ATII-injected rats. ATII significantly increased the plasma Epo concentration from 1.4 ± 0.2 to 2.9 ± 0.3 mIU/mL (* *p* < 0.05). (**b**,**c**) ATII significantly increased Epo mRNA expression in the renal cortex (13.2 ± 4.7 and 11.4 ± 0.6 times increase by A1 and A10, respectively) (**b**), and HIF2α mRNA expression was also stimulated by ATII (9.6 ± 3.7 and 8.6 ± 2.6 times increase by A1 and A10, respectively) (**c**). (**d**–**f**) In microdissected nephron segments, ATII increased Epo mRNA expression in PCT (**d**) and MAL (**f**). HIF1α mRNA was also increased by ATII in PCT (**e**). C, control; A1, ATII 1 mg/kg; A10, ATII 10 mg/kg; ATII9, AT2 10^−9^ M; ATII6, ATII 10^−6^ M. * *p* < 0.05, ** *p* < 0.01 by the *t*-test or a non-parametric analysis using the Kruskal–Wallis test and multiple comparisons by the Dunnett or Shirley–Williams’ test.

**Figure 2 molecules-26-05399-f002:**
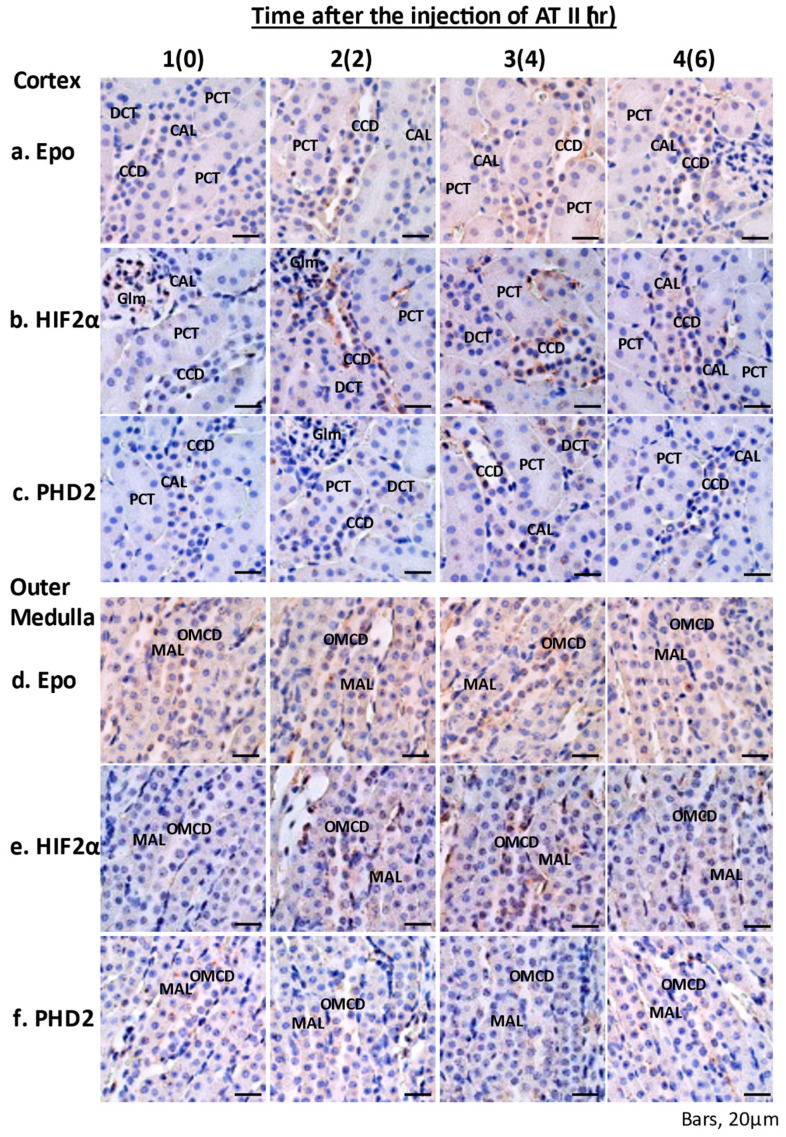
Effects of ATII on Epo mRNA expression evaluated by ISH. In situ hybridization was used to analyze the effects of ATII on Epo, HIF2α, and PHD2 mRNA expression in mouse kidneys. Epo mRNA expression was observed in cortical nephrons under basal conditions (1-**a**,**d**). ATII stimulated Epo mRNA expression in PCT, MAL, CCD, and OMCD at 2–6 h (2~4-**a**,**d**) and HIF2α mRNA expression in most nephron segments and peritubular cells of both the cortex and the medulla at 2–6 h (2~4-**b**,**e**). In contrast, ATII stimulated PHD2 mRNA expression in MAL, CCD, and OMCD at 4–6 h (3~4-**c**,**f**). (**a**–**c**) Renal cortex and (**d**–**f**) renal outer medulla. The numbers 1, 2, 3, and 4 show 0, 2, 4, and 6 h after the injection of ATII, respectively.

**Figure 3 molecules-26-05399-f003:**
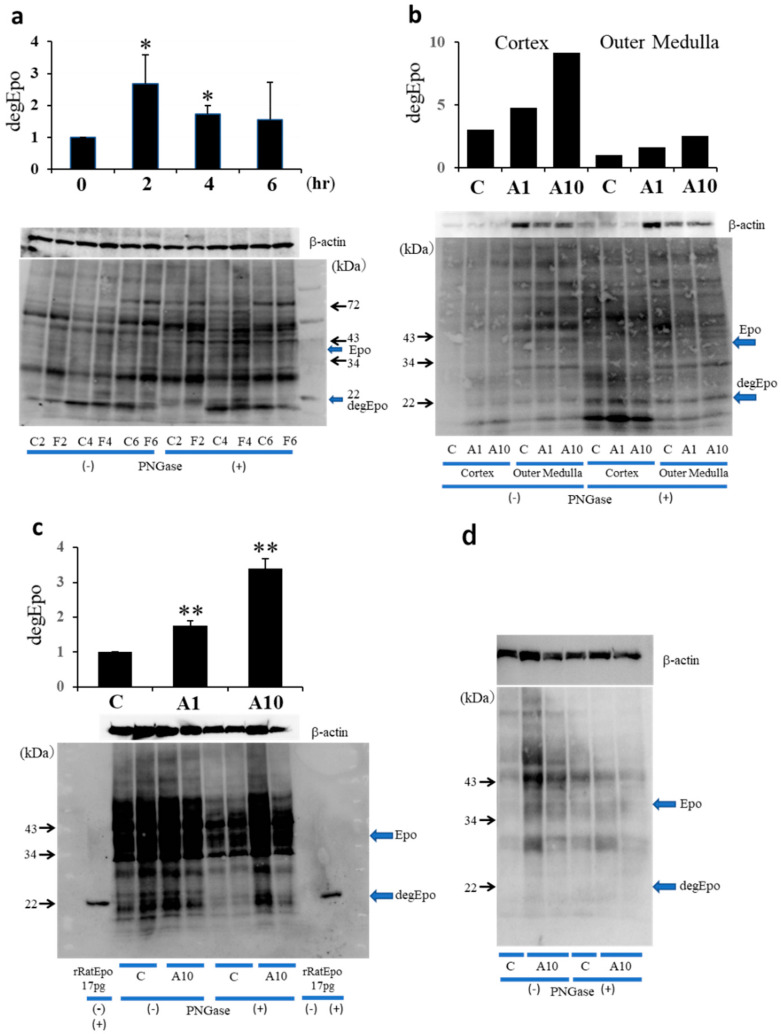
Effects of fludrocortisone and ATII on Epo protein expression in the kidney and liver. (**a**) Effects of fludrocortisone on Epo production in the kidney. Fludrocortisone increased deglycosylated Epo protein expression at 2 and 4 h after the injection. The levels returned to basal values after 6 h. (**b**) ATII dose-dependently increased deglycosylated Epo protein expression in the renal cortex and outer medulla. The effects were larger in the renal cortex than in the renal outer medulla. (**c**) ATII increased deglycosylated Epo production in the renal cortex by 1.6- and 3.4-fold by 1 and 10 mg/kg, respectively. (**d**) In contrast, deglycosylated Epo production was not observed in control and ATII-injected livers. Recombinant rat Epo is shown as the control in Figure 3c. Glycosylated Epo and deglycosylated rat Epo were used in lane 1. Glycosylated Epo and deglycosylated Epo were used in lanes 11 and 12, respectively. * *p* < 0.05, ** *p* < 0.01 by a non-parametric analysis using the Kruskal–Wallis test and multiple comparisons by the Shirley–Williams’ test.

**Figure 4 molecules-26-05399-f004:**
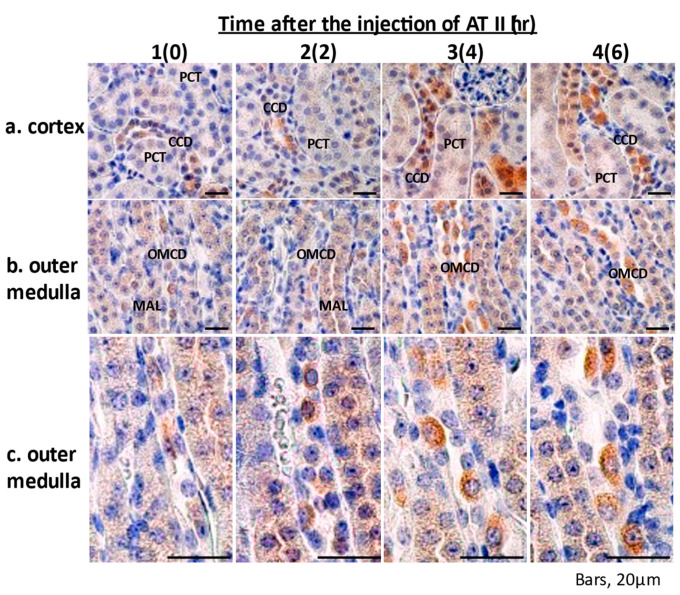
Immunohistochemical study of the effects of ATII on Epo production by the kidney. Under basal conditions, Epo staining was found from the proximal tubules to the collecting ducts (proximal tubules < thick ascending limbs < collecting ducts) (1-**a**–**c**). ATII increased Epo production in MAL, CCD, and OMCD (3~4-**a**–**c**), particularly in the type A intercalated cells, after 4–6 h (brown cells in 3~4-**a**–**c**). The renal outer medulla is shown at a different magnification. The numbers 1, 2, 3, and 4 show 0, 2, 4, and 6 h after the injection of ATII, respectively.

## Data Availability

The data presented in this study are openly available.

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
