# Peer review of "Effects of Angiotensin II on Erythropoietin Production in the Kidney and Liver"

_molecules, 2021, doi:10.3390/molecules26175399_

Round 1

Reviewer 1 Report

Dear Editor,

Thank you for asking me for the manuscript review.

The manuscript entitled “Effects of Angiotensin II on erythropoietin production in the kidney and liver” is interesting. Yasuoka et al. performed immunohistochemistry to detect erythropoietin production in the kidney and liver using deglycosylation-coupled immunoblotting. They found that angiotensinogen II stimulated erythropoietin production in the kidney. Although they showed that angiotensinogen II plays an important role in erythropoietin production, angiotensinogen II can change many pathways in kidney tissues. Before they conclude that angiotensinogen II may directly regulate the expression levels of erythropoietin, I encourage the authors do some experiments to confirm the direct effects of angiotensinogen II on the production of erythropoietin.

  1. Would you please check the effects of angiotensinogen II inhibitor on the production of erythropoietin in the cell lines
  2. Please check the effects of angiotensinogen II on promoter assay using erythropoietin promoter

These experiments will improve the quality of the manuscript.

Author Response

Answers to Reviewer 1

   We thank Reviewer 1 for his (or her) useful comments.

   As Reviewer 1 pointed out, angiotensin II has many pathways. So, the effects of angiotensin II on Epo production could be direct or indirect. Jelkmann suggested two possibilities for the mechanisms of angiotensin II-induced Epo production in his review (Ref. 4). One is a direct effect. The other is an indirect effect as a growth factor for the myeloid erythrocytic progenitors.  Kim et al. clearly reported that angiotensin II stimulated Epo production via AT1R-mediated Egr-1 activation by p21Ras-mitogen-activated protein kinase/ERK kinase-ERK/2 in human renal 786-O cells (Ref. 12 and 13). In our experiments, the effects of angiotensin II on Epo and HIF2α mRNA expression were examined by two ways. The effects of angiotensin II on Epo and HIF2α mRNA expression in renal cortex were examined 2 hrs after the injection angiotensin II to the rat. So, this effect could be a direct or indirect one. In contrast, the effects of angiotensin II on Epo and HIF2α mRNA expression in PCT and MAL were examined using microdissected PCT and MAL. PCT and MAL were incubated with or without angiotensin II for 2 hrs. Therefore, the effects of angiotensin II on Epo and HIF2α mRNA expression is a direct effect.

Answer to comments 1 and 2:

  We also thank Reviewer 1 for his (or her) suggestion of additional experiments. It is a good idea to examine the effects of an angiotensinogen II inhibitor on the production of erythropoietin in the cell lines and the effects of angiotensinogen II on promoter activity using erythropoietin promoter. The examination of the effects of an inhibitor on Epo production and promoter activity is useful to know the mechanisms of the stimulation of Epo production by angiotensin II, but the mechanisms examined in a cell line may be different from those in rat kidney. Regarding the genetic role of angiotensinogen on Epo production, Kato et al. reported nice studies (Ref 14,15). They showed that transgenic mice having renin and angiotensinogen gene exhibit erythrocytosis. They also reported that knockout mice lacking renin and angiotensinogen exhibit anemia that was rescued by ATII infusion, suggesting that a genetic pathway from angiotensinogen to ATII has a key role for ATII-induced Epo production. The suggested experiments would be future studies.

   We sincerely thank Reviewer 1 for his (or her) useful comments.

Reviewer 2 Report

It’s a good idea to look at the expression of erythropoietin. The authors use deglycosylation-coupled Western blotting to detect the epo and investigated the effects of angiotensin II (ATII) on Epo production in the kidney. The authors found that ATII stimulated plasma Epo concentration, Epo, HIF2, and PHD2 mRNA expression in nephron segments in renal cortex and outer medulla, and Epo protein expression in renal cortex. By using In situ hybridization and immunohistochemistry revealed that ATII stimulates Epo mRNA and protein expression not only in proximal tubules by also in collecting ducts, especially in intercalated cells. I would like to ask some questions about these findings and it’d be great if authors can address them.

  1. I was wondering how sensitive of the deglycosylation-coupled Western blotting to detect the epo? Any false or negative results? How you evaluated them?
  2. Regarding the expression cells, is any anatomical markers could be used and identified?
  3. The association of RAS and epo expression has been documented and established. It’d be great if authors can present more novel or further underlying mechanisms. 

Author Response

Answers to Reviewer 2

   We thank Reviewer 2 for his (or her) useful comments.

The answer to comment 1:

   The results of the Western blot were statistically evaluated.  The following data are used for Figure 3b. The expression of Epo protein was normalized to control and statistical significance was examined by non-parametric analysis using the Kruskal-Wallis test and multiple comparison by the Shirley-Williams test.

      Control          A1          A10

       1              1.57          3.03

       1              2.22          2.19

       1              0.59          4.82

       1              1.95          4.28

       1              1.22          3.44

       1              1.82         3.18

        1                            3.40

         1                            2.84

The answer to comment 2:

    In our previous paper, we examined the Epo producing cells using in situ hybridization and double staining with AQP3 (marker of principal cell) and AE1 (marker of a-intercalated cell) (Ref. 19). We found that Epo is produced by the cells that do not express either AQP3 nor AE1, suggesting that Epo is produced in ß (or non-α) intercalated cells.

The answer to comment 3:

   As you pointed out, the association between RAS and Epo expression has been established. So, we added the references and the description of the mechanisms of Epo production by angiotensin II (Ref. 4, 12 and 13). Kim et al. clearly reported that angiotensin II stimulated Epo production via AT1R-mediated Egr-1 activation by p21Ras-mitogen-activated protein kinase/ERK kinase-ERK/2 in human renal 786-O cells. Jelkmann suggested indirect effects of angiotensin II as a growth factor for the myeloid erythrocytic progenitors. As we mentioned, the role of a genetic pathway from angiotensinogen to ATII was well examined by Kato et al. (Ref. 14 and 15). They showed that transgenic mice having renin and angiotensinogen gene exhibit erythrocytosis. They also reported that knockout mice lacking renin and angiotensinogen exhibit anemia that was rescued by ATII infusion, suggesting that a genetic pathway from angiotensinogen to ATII has a key role for ATII-induced Epo production. The mechanisms of Epo production in severe hypoxic and in basal conditions may be different but some steps such as the stimulation of HIF2α could be the same. The ability of the production of Epo would be different between renal interstitial cells and nephrons.

   We sincerely thank Reviewer 2 for his (or her) useful comments.

Round 2

Reviewer 1 Report

The author well addressed the questions we issued. I recommend the manuscript publication in Molecules. 

Author Response

We will use your comments for our future studies. We thank Reviewer 1 for his (or her) useful comments.

Reviewer 2 Report

It’d be very useful to use different methods to confirm your hypothesis. A detail molecular studies would help us understanding the mechanisms. I would recommend to use some high techs to prove the study level.

Author Response

The Answer to Reviewer 2

   We thank Reviewer 2 for his (or her) useful comments.

   We used real time PCR, deglycosylation-coupled Western blot, in situ hybridization (ISH) and immunohistochemistry (IHC) in our current study.  Our study is the first one that shows the localization of angiotensin II-induced Epo production in the kidney by ISH and IHC. We have improved the sensitivity of ISH. We have spent several years finding good antibodies for IHC and Western blot. The antibodies for Western blot and IHC are different as we have shown in the Methods. We have reported that sc-5290 was more specific than the WADA-recommended clone AE7A5 for Western blot by use of deglycosylation (Ref. 5). Glycosylation of Epo results in an increase of the detection limit by 10-100 times. Since basal production of Epo by the kidney is very small and the effects of angiotensin II on Epo production is small compared with that of hypoxia, the detection of such small changes in rat kidney by Western blot became possible only by the use of deglycosylation.  Although we used traditionally established techniques, lots of improvements were required to perform our studies.

Technically high studies such as promoter assay is quite useful to understand the mechanisms of angiotensin II-induced Epo production. However, such studies need cell lines and the results obtained in cell lines might be different from those in rat kidney.  As you have mentioned in the previous comments, the effects of angiotensin II on Epo production have been established.  The investigation using high tec studies would be a future study.

Our re-revised manuscript was checked by one of our authors, Dr. Jeff M. Sands, Professor of Medicine, Emory University School of Medicine.

List of changes

Line 31-32: “are known to” was deleted

Line 33: “increase of serum” was changed to” increase in the serum”

Line 34: “people have not experienced” was changed to” people do not experience”

Line 37: “Epo in kidney” was changed to “Epo protein in the kidney”. “PNGase shifted” was changed to “PNGase shifts”.

Line 38: “The broad band of Epo” was changed to “The broad Epo band”.

Line 42: “are known to” was deleted.

Line 43-48: Previously added (possible mechanisms of angiotensin II-induced Epo production)

Line 48: “a key role for ATII-induced” was changed to “a key role in ATII-induced”.

Line 56: “Administration of ATII” was changed to “ATII administration”.

Line 58: “proximal convoluted tubule”

was changed to “proximal convoluted tubules”.

Line 59: “medullary thick ascending tubule” was changed to “medullary thick ascending limb”.

Line 79: “after the injection” was changed to “after injection”.

Line 80-81: “The effect” was changed to “The effects”.

Line 84: “n=8-9” was changed to “n=6-8”.

Fig.1 was changed.

Line 103: “dect”was previously changed to “detect”

Line 105: “in renal cortex” was changed to “in the renal cortex”.

Line 106-107: proximal convoluted tubules and medullary thick ascending limbs were deleted.

Line 108: “in site hybridization” was previously changed to “in situ hybridization (ISH).

Line 112: “whici is compatible wihe the previous reports” was previously changed to “which is compatible with previous reports”.

Line 117-118: “Our study is the first one that shows the intrarenal localization of ATII-induced Epo production by ISH and IHC” was added.

Line 122-132: Previously added as possible mechanisms of angiotensin II-induced Epo production.

Line 126: “was well examined” was changed to “was examined”.

Line 129: “genetic pathway” was changed to “a genetic pathway”. “a key role for” was changed to “a key role in”.

Line 130-131: “in severe hypoxia and in basal condition” was changed to “in severe hypoxia and basal condition”.

Line 131: “stimulatoin” was changed to “stimulation”. “be same” was changed to “be the same”. “The ability of the production” was changed to “The ability to produce”.

Line 144-145: proximal convoluted tubules and medullary thick ascending limbs were deleted.

Line 189-190: “4% Nonidet P-40 and 8% 2-mercaptoethanol” was changed to “2% Nonidet P-40 and 4% 2-mercaptoethanol”.

Line 212-213; “in basal condition” was changed to “under basal conditions”

We sincerely thank Reviewer 2 for his (or her) useful comments.